# Histone Deacetylase Inhibitors Impair Vasculogenic Mimicry from Glioblastoma Cells

**DOI:** 10.3390/cancers11060747

**Published:** 2019-05-29

**Authors:** Olga Pastorino, Maria Teresa Gentile, Alessandro Mancini, Nunzio Del Gaudio, Antonella Di Costanzo, Adriana Bajetto, Paola Franco, Lucia Altucci, Tullio Florio, Maria Patrizia Stoppelli, Luca Colucci-D’Amato

**Affiliations:** 1Department of Environmental, Biological and Pharmaceutical Sciences and Technologies, University of Campania “Luigi Vanvitelli”, 81100 Caserta, Italy; olga.pastorino@gmail.com (O.P.); mariateresa.gentile@unicampania.it (M.T.G.); 2Department of Translational Medical Sciences, University of Campania “Luigi Vanvitelli”, 80138 Naples, Italy; alessandro.mancini@unicampania.it; 3BioUp Sagl, 6900 Lugano, Switzerland; 4Department of Precision Medicine, University of Campania “Luigi Vanvitelli”, 80138 Naples, Italy; nunzio.delgaudio@unicampania.it (N.D.G.); antonella.dicostanzo@unicampania.it (A.D.C.); lucia.altucci@unicampania.it (L.A.); 5Pharmacology Unit, Department of Internal Medicine and Centre of Excellence for Biomedical Research (CEBR), University of Genova, 16132 Genova, Italy; adriana.bajetto@unige.it (A.B.); tullio.florio@unige.it (T.F.); 6IRCCS Ospedale Policlinico San Martino, 16132 Genova, Italy; 7Institute of Genetics and Biophysics “A. Buzzati Traverso” (IGB-ABT), National Research Council of Italy, 80131 Naples, Italy; paola.franco@igb.cnr.it (P.F.); mpatrizia.stoppelli@igb.cnr.it (M.P.S.); 8InterUniversity Center for Research in Neurosciences (CIRN), 80131 Naples, Italy

**Keywords:** angiogenesis, glioblastoma, cancer stem cells, cell migration, cell invasion, HDAC, vasculogenic mimicry

## Abstract

Glioblastoma (GBM), a high-grade glioma (WHO grade IV), is the most aggressive form of brain cancer. Available treatment options for GBM involve a combination of surgery, radiation and chemotherapy but result in a poor survival outcome. GBM is a high-vascularized tumor and antiangiogenic drugs are widely used in GBM therapy as adjuvants to control abnormal vasculature. Vasculogenic mimicry occurs in GBM as an alternative vascularization mechanism, providing a means whereby GBM can escape anti-angiogenic therapies. Here, using an in vitro tube formation assay on Matrigel^®^, we evaluated the ability of different histone deacetylase inhibitors (HDACis) to interfere with vasculogenic mimicry. We found that vorinostat (SAHA) and MC1568 inhibit tube formation by rat glioma C6 cells. Moreover, at sublethal doses for GBM cells, SAHA, trichostatin A (TSA), entinostat (MS275), and MC1568 significantly decrease tube formation by U87MG and by patient-derived human GBM cancer stem cells (CSCs). The reduced migration and invasion of HDACis-treated U87 cells, at least in part, may account for the inhibition of tube formation. In conclusion, our results indicate that HDACis are promising candidates for blocking vascular mimicry in GBM.

## 1. Introduction

Glioblastoma (GBM) is the most common and lethal among nervous system tumors, classified as grade IV glioma by the World Health Organization (WHO) [1,2,3,4]. The major pathological features of GBM are uncontrolled cellular proliferation, hypervascularization, necrosis, de-differentiation, genetic instability and aggressive invasiveness [5]. Current available therapies consist of surgical resection followed by radiotherapy and chemotherapy. Despite this combination of treatments, prognosis is extremely poor with a median survival of about 14-months, and average 5-year survival rate of about 5% [4,6]. Temozolomide, the gold standard cytotoxic drug for gliomas, and cisplatin are two DNA alkylating agents largely used in GBM chemotherapy [7,8]. However, these drugs cause cognitive impairment due to normal brain cell damage [9,10], and tumors treated with these drugs develop chemo-resistance, largely due to high *O*^6^-methylguanine-DNA methyltransferase (MGMT) levels, to inactivation of mismatch repair enzymes MLH1 and MSH29 and over-expression of multidrug resistance proteins [10]. Moreover, it has been suggested that GBM resistance to radio- and chemotherapy and the subsequent tumor recurrence after surgical resection is mainly due to cancer stem cell (CSC) subpopulations, which promote tumor initiation [11,12,13,14], making this cancer one of the most difficult to treat [10,14,15,16,17,18]. Another hallmark of GBM is high vascularization. Neo-angiogenesis is the most studied mechanism of vascularization in tumors and refers to sprouting of new blood vessels from pre-existing ones [19,20]. Vascular endothelial cell growth factor (VEGF), interacting with its receptors (VEGFRs), is a strong stimulator of endothelial cell (ECs) proliferation and mobility during angiogenesis [21] and high levels of VEGF are detected in the tumor microenvironment [22]. However, vascularization in tumors is a more complex process involving several other mechanisms [19] such as recruitment of endothelial progenitor cells (EPCs) [23], vascular co-option of vessels [24,25], intussusceptive angiogenesis [26,27] and vasculogenic mimicry [28]. An important role in GBM vascularization is played by GBM CSCs, which promote angiogenesis through the release of VEGF [29] but, also, can participate directly in the formation of tumor cell-lined blood vessels developing a vascular phenotype [28,29,30,31].

Indeed, the ability of aggressive and genetically dysregulated tumor cells to form fluid-conducing network [28] is well known. In the last decade, Sanson et al. have described the occurrence of vasculogenic mimicry in GBM showing, by means of refined immunohistochemistry studies, the presence of cell-lined blood vessels inside the tumor, with a functional basement membrane but independent from normal ECs and mural cells [32]. In this and other studies, it has been reported that GBM CSCs can transdifferentiate into smooth muscle-like cells expressing mural markers. These cells may contribute to tumor cell-lined vessel’s wall formation, providing contractile properties [32,33]. On the other hand, it has been also described that GMB CSCs transdifferentiate into functional endothelial-like cells expressing endothelial markers (CD31 and/or CD144) [30,31,34]. Altogether, such evidence demonstrates the high plasticity of GBM CSCs and the possibility of GBM cells to form, completely de novo, the whole structure of blood vessels [32]. The transdifferentiation of GBM CSCs into endothelial-like cells may involve an epithelial-mesenchymal transition (EMT) that is crucial in vasculogenic mimicry process in GBM [35]. The molecular mechanisms underlying tumor blood vessels formation are promising target of pharmacological research aimed at reducing tumor vascular irrigation and preventing oxygenation and nutrient supply to the tumor. Anti-angiogenic drugs, mostly affecting VEGF-VEGFR pathway, cause dramatic tumor size reduction and are largely used in GBM therapy as adjuvants to control abnormal vasculature. However, the benefits are transient since GBM rapidly show resistance to anti-angiogenic therapies (AATs) in prolonged treatment [35,36,37,38] through the activation of alternative vascularization pathways [38]. Moreover, it has been reported that hypoxia associated with AAT may induce vasculogenic mimicry [39] that represents a mechanism whereby GBM can escape anti-angiogenic therapies. Therefore, vasculogenic mimicry become an interesting therapeutic target in GBM therapy. As in other human cancers, epigenetic mechanisms are recognized as important factors contributing to GBM pathogenesis [40,41]. In particular, an essential role has been attributed to histone acetylation during tumorigenesis. Acetylation of histones depends on the balance between the activities of two classes of antagonizing histone-modifying enzymes: histone deacetylases (HDACs) and histone acetyltransferases (HATs) [42]. HDACs regulate gene expression modifying histones acetylation but also interacting directly with transcription factors as co-repressor or activator of genes expression [41]. Indeed, HDACs’ substrates are not only histones, but also non-histone proteins, which regulate important functions in cell proliferation, death and cell differentiation [43,44,45]. In fact, HDACs precede histone proteins in phylogenetic evolution, suggesting non-histone proteins as primary substrates for HDACs activity [46]. Altered expression and/or function of HDACs could play an important role in tumor initiation and progression. Different natural and synthetic compounds inhibiting HDACs activity (HDAC inhibitors, HDACis) were largely tested in different cancers, with promising results in affecting proliferation, invasion, angiogenesis and resistance to apoptosis [41]. Interesting in vitro and in vivo results on antitumor activity of HDACis were demonstrated also in GBM and different HDACis entered in clinical trials for this disease. In particular, vorinostat (SAHA), romidepsin and valproic acid (VPA), have entered clinical trials for GBM in combination with traditional chemotherapy [46]. Moreover, HDACis are known to show neuroprotective effects in vitro in several models of neurodegenerative disorders such as Alzheimer’s and Parkinson’s disease [47]. Here, we evaluated whether different histone deacetylase inhibitors, with different selectivity for HDAC classes, are able to impair GBM cell line and GBM CSCs capability to form tube-like structure on extracellular matrix (ECM) in vitro, as model of vasculogenic mimicry. In particular, we tested vorinostat (SAHA) and trichostatin A (TSA) as inhibitors of class I and II HDACs, entinostat (MS275) as selective inhibitor of class I HDACs (HDAC 1 and 3) and MC1568 as selective HDAC class II inhibitor.

## 2. Results

### 2.1. U87MG and C6 Cells Are Able to form Tube-Like Structures on Extracellular Matrix

Highly aggressive tumor cells are able to form vessel-like structures and patterned networks because of their plasticity [30,31,32,34]. In order to clarify whether C6 and U87MG glioma cell lines may undergo a vasculogenic switch in three-dimensional cultures and, therefore, are suitable models to study vasculogenic mimicry in vitro, we performed tube formation assays on a gelled basement membrane. When C6 rat glioma cells and U87MG human glioblastoma cell line are seeded onto polystyrene dishes, they grow adherent to the surface without forming any particular structure (Figure 1A). In contrast, when cultivated in 3D-Matrigel^®^ (ECM), under serum-free conditions, they form clear-cut tubes developing from elongated cell bodies that connect to each other and form polygonal networks similar to those formed by HUVECs (Figure 1A). Upon tube formation, U87MG cells are negative to the endothelial marker CD31, but they express VE-Cadherin, a hallmark of vasculogenic mimicry [48] (Figure 1B). Thus, these cell lines can be appropriate tools to evaluate in vitro the potential anti-vasculogenic effect of HDACis. Among HDACis, we tested SAHA and TSA, inhibitors of class I and II HDACs, MS275, selective inhibitor of class I HDACs (HDAC 1 and 3) and MC1568, selective HDACs class II inhibitor (Table 1).

### 2.2. HDACi Reduce Tube Formation by C6 and U87 Glioma Cell Lines

When administered at 50 nM, HDACis reduce tube formation by rat glioma C6 cells. As shown in Figure 2A,B, in C6 cells SAHA and MC1568 reduce vascular tube formation by 60% and 68%, respectively, as compared to control conditions. TSA and MS275 did not show any significant effect to prevent tube formation of C6 cells. It is noteworthy that no significant difference is detected in treated groups as compared to the control group in MTT assay, after 24 h of treatment at the same concentration (Figure 2C).

Similar testing was performed on U87MG human glioblastoma cells exposed to SAHA, TSA, MS275 and MC1568 at the concentrations indicated in the figure legend (Figure 3), before being seeded on Matrigel^®^. When administered at 50 nM, HDACis strongly inhibit tube formation of U87MG cells. In particular, TSA, MS275, SAHA and MC1568 reduce by 80%, 87%, 90% and 94% tube formation, respectively, as compared to control conditions (Figure 3A,B). Cell viability was unchanged by treatments as shown by MTT assay (Figure 3C).

### 2.3. TSA, MS275, MC1568 HDAC Inhibitors Decrease Cell Motility of Human U87MG Cells

Cell motility is an essential prerequisite for tube formation [49], thus we first evaluated the effects of HDACis on migration of U87MG cells by means of Boyden chamber migration assay and xCELLigence^®^ Real-Time Cell Analysis (ACEA Bioscences, Inc. San Diego, CA, USA). The experiments were carried out at 50 nM and the effects were evaluated after 3h in Boyden chamber assay and after 24h in xCELLigence^®^ Real-Time Cell Analysis (Figure 4B,C). Before performing our experiments, activity of HDACis representative of all the groups, was checked by analysing the acetylated H3K9K14 in histonic extracts from U87MG exposed to a specific HDACi at used concentrations. H4 was used as loading control (Figure 4A).

Results show that in presence of TSA, MS275 and MC1568 directional cell migration of U87 MG cells towards FBS significantly decreases. In Boyden chamber assays MS275, MC1568, and TSA decrease to 47%, 45% and 62% cell migration, respectively, as compared to directional cell migration towards FBS in the absence of inhibitors (taken as 100%). It is noteworthy that all inhibitors are able to completely inhibit the chemotactic response to FBS, reducing migration rate to the basal level of random migration (migration of untreated cells in the absence of chemo-attractant) for TSA and below for MS275 and MC1568 (Figure 4B). Also xCELLigence^®^ Real-Time Cell Analysis show that HDACis MS275, MC1568 and TSA reduce directional cell migration to 38%, 50% and 60%, respectively as compared to controls (taken as 100%) (Figure 4C).

Moreover, we tested the capability of MS275, the most effective compound in the migration assays, to impair U87MG cells ability to invade ECM. In Boyden chamber invasion assays, after 20 h, MS275 decreased cell invasion toward FBS to 44%, as compared to the control (DMSO, taken as 100%) (Figure 5). It is of note that this compound is able to inhibit completely the chemotactic response to FBS and the invasion rate become comparable to that of random invasion (invasion of untreated cells in the absence of chemo-attractant). Altogether, these results lead us to investigate whether HDACis could be good candidates to inhibit tube-like structure formation of glioma cells.

### 2.4. HDACis Inhibit Vasculogenic Mimicry of Human GBM CSCs

In CSCs (GBM1 and GBM2), isolated from two different human GBMs, MC1568, SAHA and MS275, reduce cell ability to form vascular tube, after 24 h of treatment, with a higher efficacy in GBM2 cells (Figure 6A,B). Again, no statistically significant difference is detected in treated groups in cell viability, after 24 h of treatment (Figure 6C), although the effective concentrations able to impair CSCs vasculogenic mimicry are higher than those required using GBM cell lines (MC1568: 100 nM; SAHA and MS275: 5 µM).

## 3. Discussion

Here we show that in vitro tube-like structure formation by GBM cell lines and GBM CSCs can be impaired by HDACis. For this purpose, we performed tube formation assay on Matrigel^®^ (Corning^®^, Bedford, MA, USA), a reliable in vitro assay useful to evaluate, in quantitative manner, the cell ability to form tube-like structures. Even if in vivo models could give more information on the angiogenesis processes, this assay has been widely used to study both normal and alternative angiogenesis, including vasculogenic mimicry [32,49,50] since it investigates at the same time cell proliferation, adhesion, migration and protease activity, all crucial events in tube formation and thus in vasculogenic mimicry [49,50]. Moreover, this assay, being highly reproducible, is indicated to identify inhibitors or stimulators of tube formation and to study the relative signal transduction more easily than in vivo models. Indeed, GBM cells, plated on a gelled basement matrix in the absence of serum, are able to form tube-like structures similar to those formed by human endothelial primary cells (HUVEC). Differently from HUVECs, tube forming-U87MG are positive to VE-cadherin and negative to CD31. The expression of VE-cadherin but not of CD31 is regarded as a well-established marker of tumor cell-lined vasculogenic mimicry [48]. In fact, it has been reported that the genetic inhibition of VE-cadherin expression inhibits alternative vascularization mechanisms indicating that VE-cadherin may be associated with vasculogenic mimicry [48,51]. The process of vasculogenic mimicry in vivo depends on the ability of highly genetically dysregulated tumor cells to form tubular structures mimicking endothelial vessels [30,32]. The clear-cut capability of U87MG and C6 cells, as well as human GBM CSCs, to form tube-like structures make these cells good models for our in vitro studies on vasculogenic mimicry.

Cellular migration and invasion are central hubs in the tube formation process. In particular, cellular invasion depends on migration through the dense extracellular matrix barrier and requires, in addition to adhesion and migration, the proteolysis of ECM components [52]. For these reasons, we preliminary assessed the effects of TSA, MS275 and MC1568 on U87MG directional cell migration using two different transwell migration assay systems, such as the Boyden chamber and xCELLigence^®^ assays. MS275 was the most effective HDACis in decreasing cell migration and, thus, was tested in invasion Boyden chamber assay. We found that the tested HDACis significantly inhibit FBS-dependent directional cell migration and invasion, suggesting that they affect important components underlying vasculogenic mimicry process.

We found that all the HDACis tested are able to impair tube formation in human GBM U87MG cell line and in human GBM CSCs. In rat glioma C6 cells, at the same concentrations used in U87MG, only SAHA and MC1568 interfere with vessels formation, whereas TSA and MS275 do not display activity to prevent tube-like structures. However, it is possible that in C6 cells TSA and MS275 need higher concentrations to be effective. These findings may suggest that these HDACis could discriminate among different species (human vs. rat) or among tumors harboring different genetic mutations. Furthermore, HDACis affect the ability of CSCs, derived from two differently classified human tumors (Table 1), to form vascular structures. It is worth to note that the HDACis concentrations required to impair this process in CSCs are higher, reflecting the well-known higher resistance of CSCs to pharmacological treatments as in the case of temozolomide resistance [10,16,17,18]. However, the concentrations used in our experimental settings did not display cell toxicity, as shown by MTT assay.

The finding that all the HDACis, with different specificity for HDAC classes (Table 1), are active to prevent tube formation by human GBM suggests the involvement of a common or convergent mechanism of action, worth of further investigation. Here we show that mechanisms involved in cell migration and invasion are impaired by HDACis. We also found that pre-treating U87MG with HDACis causes an increase of early adhesion, thus, corroborating our data on cell motility inhibition and suggesting that changes in cell adhesion can occur during tube formation in treated cells. Moreover, it is possible that HDACis are able to induce a differentiation program preventing cellular transdifferentiation underlining tube formation process in GBM cells. Indeed, it has been reported that many HDACis are able to interfere with differentiation status in GBM cells [45,53]. For instance, preclinical studies demonstrate that, in several human GBM cell lines, low concentrations of TSA increase neuronal markers expression and decrease the expression of vimentin and nestin, a neuro-epithelial stem cells marker. This evidence suggests that neural differentiation can be induced by TSA in GBM cells [53]. Moreover, stimulation of neurospheres derived from GBM with etinostat (MS275) or TSA causes the expression of the Delta/Notch-like epidermal growth factor-related receptor (DNER), which inhibits the neurospheres growth and induces their differentiation both in vivo and in vitro [54]. Furthermore, MS275, the class I HDAC inhibitor, is more effective than MC1568, the class II HDAC inhibitor, in enhancing the hTPH2 promoter activity [55] by the transcription factor NRSF, crucial in neural cell differentiation [56].

Further studies are underway to confirm the effect on a larger number of cell lines and CSCs and to identify the molecular mechanisms of HDACis in reducing vasculogenic mimicry in GBM cells. In particular, we are investigating the molecular effects of these compounds in inhibiting cell motility and their interference on the epithelial-mesenchymal transition during tube formation by GBM cells.

## 4. Materials and Methods

### 4.1. Cell Lines Cultures

Human glioblastoma U87MG cell line was cultured in DMEM (Gibco-Life Technologies, Paisley, UK) containing 10% fetal bovine serum (FBS) (Gibco-Life Technologies). Rat glioma C6 cell line was cultured in MEM/F12 (1:1) (Gibco-Life Technologies) 10% FBS as previously described [57]. The cell lines were obtained from the Bank of Biological Material of the Interlab Cell Line Collection (National Institute for Cancer Research, Genova, Italy). Human umbilical vein endothelial cells (HUVECs) were obtained from ATCC and maintained in endothelial cell growth medium: EBM2 medium (Lonza Srl, Milano, Italy) supplemented with 1 μg/mL hydrocortisone and 1 ng/mL epidermal growth factor and 10% FBS. All media were supplemented with 100 U/mL of penicillin/streptomycin (Lonza). Cells were incubated in a humidified tissue culture incubator at 37 °C, 5% CO_2_, 95% air atmosphere.

### 4.2. Human GBM Specimens and CSC Cultures

Two tumor specimens were obtained from Neurosurgery Dept. of IRCCS Ospedale Policlinico San Martino (Genova, Italy), after Institutional Ethical Committee approval of the informed consent provided to patients and the ex vivo human sample study (ethic code 17/12, approved 14 September 2012). Both samples were diagnosed as primary WHO grade IV GBM and individual characteristics are reported in Table 2. Patients underwent first-time surgery and did not receive prior radio/chemo-therapy. After resection, tumor specimens were immediately processed to isolate single cells by mechanical dissociation and cell suspension was filtered through a 40 µm strainer (BD Biosciences, Buccinasco, Milano, Italy) to remove aggregates, and cultured as previously described [58]. Human CSCs obtained were grown in stem cell-permissive medium DMEM-F12/Neurobasal (1:1), supplemented with 1% B27 (Life Technologies, Carlsbad, CA, USA), 2 mM L-glutamine (Lonza), 1% penicillin/streptomycin (Lonza), 10 ng/mL bFGF and 20 ng/mL EGF (PeproTech, London, UK) [59]. Cells were grown as monolayer on Matrigel^®^ (BD Biosciences, Milano, Italy), experimental conditions that allow cells to retain CSCs features, as reported [60]. This culture condition was used to obtain easier evaluation of cell biology and biochemical experiments, rather than using non-adherent spheroids. In previous experiments [61], we demonstrated the tumorigenicity of both GBM CSCs cultures by intracranial inoculation of 10^4^ cells/mouse, in 6–8-week-old NOD/SCID mice (Charles River, Calco, Italy); in compliance with guidelines approved by Ethical Committee for animal use in cancer research at IRCCS Ospedale Policlinico San Martino (ethics code 17/12 approved 14 September 2012) (Genova, Italy).

### 4.3. Drug Preparation and Cells Treatment

TSA (Sigma Aldrich, Milano, Italy), Tubacin (Sigma Aldrich), MS275 (Bayer-Schering AG, Leverkusen, Germany), MC1568 (Selleckchem, Houston, TX, USA) and SAHA (Merck, Darmstadt, Germany) HDACis were dissolved in dimethylsulfoxide (DMSO) (J.T. Baker, Deventer, The Netherlands) at appropriate concentration and stored at –20 °C until further use. In the different experimental setting, cells were exposed either to HDACi or to DMSO.

### 4.4. MTT Assay

In 24-well plates, U87MG and C6 cells were plated at 1 × 10^4^ cells/well and 0.5 × 10^4^ cells/well respectively; 24 h later, the cells were treated with different HDACis at specified concentration; the cellular proliferation was measured at 24 h by Thiazolyl Blue tetrazolium bromide (MTT) assay as previously described [57]. Briefly, MTT (Invitrogen-Life Technologies, Eugene, OR, USA) 5 mg/mL PBS solution was added to wells at final concentration of 10%. After 3 h of incubation at 37 °C, was added one volume of stop solution (20% SDS in 50% dimethylformamide). After a minimum of 1 h at room temperature, the optical densities were measured at 550 nm by Microplate Reader (Synergy™ HT, Bio-Tek (Winoosky, VT, USA).

The cell viability was calculated as follows:Cell viability rate (%) = (A_test group_/A_negative control group_) × 100

### 4.5. In vitro Tube Formation Assay

For U87MG and C6, tube formation assay was performed in 96-well tissue culture plates coated using 50 µL/well of Matrigel^®^ (Corning^®^, Bedford, MA, USA) with ECM proteins concentration of 9.3 mg/mL. The matrigel was allowed to polymerize at 37 °C for 30 min. The U87MG and C6 cells were dissociated by trypsinization, washed in PBS (+Mg^2+^, +Ca^2+^) and re-suspended in serum-free medium at 2 × 10^5^ cells/mL and 1.7 × 10^5^ cells/mL respectively. To evaluate the effect of HDACis on tube formation, the cell suspensions were divided into control groups (DMSO 0.1%) and several treatment groups to receive the different HDACis at final concentration of 10 nM or 50 nM. Subsequently, 0.1 mL/well cell suspensions were plated onto the surface of Matrigel^®^ and incubated at 37 °C for 24 h or 16 h in U87MG or C6 experiments respectively. In order to perform tube formation assay on Human GBM CSC μ-Slide Angiogenesis (Ibidi, Munich, Germany) were coated with Matrigel^®^ (BD Biosciences) and allowed to polymerize at 37 °C for 30 min. 10^4^ cells were subsequently seeded on Matrigel^®^ and incubated at 37 °C for 24 h, in Endogro medium (Merck Millipore, Darmstadt, Germany) to induce endothelial transdifferentiation [61]. Following incubation, each well was analyzed directly under an inverted phase contrast microscope (Axiovert 25, Zeiss, (Oberchoken, German), tubes in each field were imaged and the number of branch sites/nodes (sites of intersection of least 3 tubes) was counted using the ImageJ.Ink software (1.51i, NIH, Bethesda, MD, USA, ). The average of branch point number from 5–6 random fields in each well was reported. HUVECs were plated on Matrigel^®^ in 96-well plates at 2 × 10^4^ cells/well, and were incubated at 37 °C for 24 h as previously described [62] as tube assay positive control.

### 4.6. Immunohistochemistry

U87MG and HUVEC cells were seeded on Matrigel^®^ for Tube Assay as previously described; tubes were fixed with 4% PFA solution for 20’ at room temperature. The fixed tubes were treated with PBS 10% FBS blocking solution for 1 h at room temperature and incubated with monoclonal I antibodies anti VE-cadherin (BioLegend Inc., San Diego, CA, USA) (1:250) or anti-CD31 (BD Biosciences Pharmingen, San Diego, CA, USA) for 3 h at room temperature. The samples were then incubated with II Ab anti mouse IgG HRP conjugated (ImmunoReagents Inc., Raleigh, NC, USA) (1:5000 in PBS) for 1 h at room temperature. Antibody positivity was revealed by peroxidase substrate DAB substrate Kit (Vector Laboratories, Burlingame, CA, USA). Cells were observed by a light inverted microscope at 10× magnification. Negative controls were incubated with anti-mouse IgG only.

### 4.7. Histone Extractin and Western blotting Analysis

1 × 10^6^ U87 MG cells were treated with 50 nM of indicated compound for 24 h. Cells were harvested and washed twice with cold 1× PBS and lysed in Triton extraction buffer (TEB: PBS containing 0.5% Triton X-100 (v/v), 2 mM PMSF, 0.02% (w/v) NaN_3_) for 10 min on ice, with gentle stirring. After a brief centrifugation at 2000 rpm at 4 °C, the supernatant was removed and the pellet was washed in half the volume of TEB and centrifuged as before. The pellet was suspended in 0.2 M HCl and acid extraction was left to proceed overnight at 4 °C on a rolling table. Next, the samples were centrifuged at 2000 rpm for 10 min at 4 °C, the supernatant was removed and protein concentration was determined using the Bradford assay. Immunoblotting was performed using acetylated histone H3 (H3K9K14) and H4 (Upstate Biotechnologies, New York, NY, USA) antibodies.

### 4.8. Boyden Chamber Migration Assay

U87MG directional cell migration has been assessed by Boyden chamber assay; 8 µm pore size filters (Whatman™ nucleopore track-etched, Sigma-Aldrich) coated with 50 µg/mL collagen type IV (Sigma-Aldrich) were employed. The cells were detached by mild trypsinization, washed in PBS, re-suspended in DMEM/0.1% bovine serum albumin (BSA) (Sigma-Aldrich) and pre-treated with the indicated HDAC inhibitors at final concentrations of 50 nM for 1 h at 37 °C. 10^5^ cells/sample were added to the upper compartment, while the chemoattractant, consisting of 10% FBS diluted in DMEM/0.1% BSA, was added to the lower compartment. After incubation at 37 °C for 3 h, the cells, able to migrate toward the lower compartment, were fixed, stained with hematoxylin and counted under microscope along fields of the filter diameter. Number of migrated cells is expressed as a percentage of the cells migrated in the absence of chemoattractant (random migration), taken as 100%. As a control, we analyzed the FBS-dependent migration of cells in the presence of DMSO, the HDACis molecules solvent. Experiments were performed in triplicate.

### 4.9. Real-Time Migration Monitoring

To monitor cell migration in real time, we used the xCELLigence Real-Time Cell Analyzer (RTCA) DP Instrument equipped with a CIM-plate 16 (ACEA Biosciences, Inc., San Diego, CA, USA). The CIM-plate 16 is a 16-well system in which each well is composed of upper and lower chambers separated by an 8-μm microporous membrane. These specially designed “trans-wells” measure capacitance on the underside of the dividing membrane (RTCA DP instrument, ACEA Biosciences, Inc.). Migration was measured as the relative impedance change (cell index) across microelectronic sensors integrated into the bottom side of the membrane. FBS (10% in DMEM) was used as a chemoattractant. U87 MG cells (3 × 10^4^) were starved in a serum-free media and treated with several HDAC inhibitors such as MS275, MC1568 and TSA at a final concentration of 50 nM. Treated and control (DMSO) cells were added in triplicates to the upper chambers and leaved at room temperature for at least 30 min to let cells settle. RTCA DP instrument was placed in a humidified incubator maintained at 37 °C with 5% CO_2_ for 24 h. The xCELLigence software was set to collect impendence data (reported as percentage of cell index) at least once every 15 min for 24 h. For quantification, the cell index at 24 h, for each HDACis treatment, was averaged from at least three independent measurements.

### 4.10. Invasion Assays

U87MG cell invasion have been assessed by Boyden chamber assay; 8 µm pore size filters were coated with 50 µg/mL collagen type IV, dried and then a second coating with 50 µg/mL Matrigel^®^ (Corning^®^) was used. Cell suspension was prepared and pre-treated as described for Boyden chamber migration assay. 0.5 × 10^5^ cells/sample were added to the upper compartment, whereas the chemoattractant, consisting of 10% FBS diluted in DMEM/0.1% BSA, was added to the lower compartment. After incubation at 37 °C for 20 h, the migrated cells were fixed, stained with hematoxylin and counted under microscope along fields of the filter diameter. Number of migrated cells is expressed as a percentage of the invaded cells in the absence of chemoattractant (random invasion), taken as 100%. As a control, we analyzed the FBS-dependent invasion of cells in the presence of DMSO, the HDACis molecules solvent.

### 4.11. Statistical Analysis

Data were expressed as mean ± SD. T test were performed and *p*-value < 0.05 was considered as statistically significant.

## 5. Conclusions

Altered HDACs expression and/or function, modifying the normal epigenetic arrangement, play an important role in tumor initiation and progression. Unlike genetic mutations, epigenetic alterations are reversible and can be targeted by specific drugs. Here, we provide substantial in vitro evidences that HDACis inhibit human GBM cells ability to form vascular tubes, the cellular ability underlying vascular mimicry process in vivo [43,44,45]. To the best of our knowledge, this is the first report showing that HDAC inhibitors alone can counteract vasculogenic mimicry in GBM. Thus, our results may open new therapeutic perspectives on the use of these compounds as agents able to target tumor neovascularization generated by vasculogenic mimicry.

## Figures and Tables

**Figure 1 cancers-11-00747-f001:**
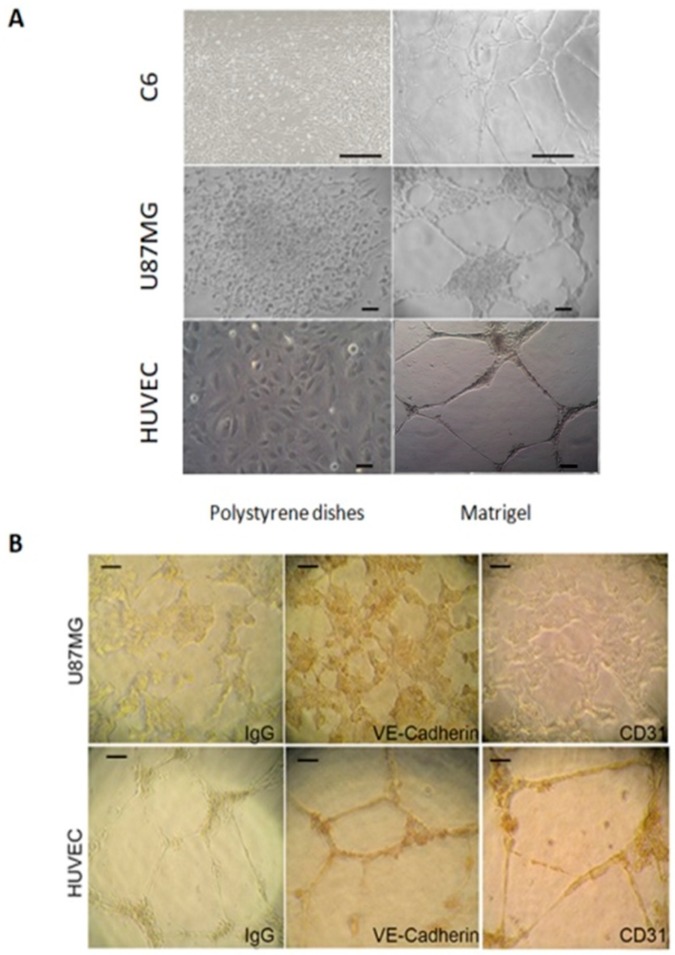
Tube formation by U87MG and C6 cells onextra cellular matrix (ECM). (**A**) Representative images of U87MG and C6 cells cultured in polystyrene dishes with or without ECM. Cells were analyzed under an inverted phase contrast microscope and imaged. 10× magnification was used for U87 and HUVEC, 5× magnification was used for C6 cells. Scale bars: 50 µm in U87MG and HUVEC images and 250 µm in C6 images. (**B**) In human GMB U87MG cell line, vascular tube formation depends on vasculogenic mimicry. Immunocytochemistry assay on tube formed by U87MG cells was employed in order to assess the positivity to VE-Cadherin, but not to CD31. Antibody positivity was revealed by peroxidase substrate DAB kit. Controls were incubated with II Ab (IgG) to exclude false positive signals. Representative images, observed by a light inverted microscopy at 10× magnification, were shown. HUVECs were used as a positive control. Scale bars: 50 µm.

**Figure 2 cancers-11-00747-f002:**
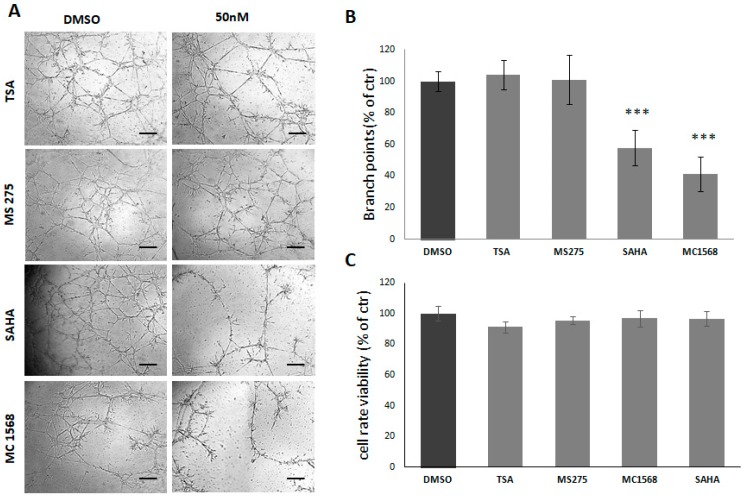
Effect of HDACis on C6 cells ability to form vascular tubes. (**A**) Representative images of vascular tubes formed by C6 cells in control group (DMSO 0.1%) and in treated group. All HDACis are used at final concentration of 50 nM. Network of tubes, in each well, were analyzed directly under an inverted microscope with 5× phase contrast and imaged. Scale bars: 250 µm. (**B**) Quantification of vascular branching. Branch point (sites of intersection of at least three tubes) number in each well was counted and expressed as a percentage of branch points formed in the control sample, taken as 100%. As shown, SAHA and MC1568 reduce C6 ability to form vascular tubes, whereas TSA and MS275 have no effect. (**C**) The cell rate viability assayed by MTT assay. At concentration of 50 nM, no differences in 24 h was detected in treated groups as compared to the control group. *** *p*-value < 0.001.

**Figure 3 cancers-11-00747-f003:**
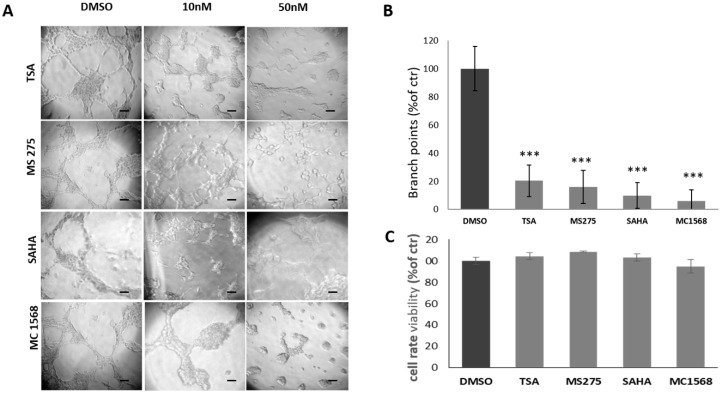
Effect of HDACis on U87MG cells ability to form vascular tubes. (**A**) Representative images of vascular tubes formed by U87MG cells in control group (DMSO 0.1%) and in treated groups with HDACis at final concentrations of 10 and 50 nM. Network of tubes, in each well, were analyzed directly under an inverted microscope with 10× phase contrast and imaged. Scale bars: 50 µm. (**B**) Quantification of vascular branching in control group and groups treated with 50 nM HDACis. Branch point (sites of intersection of at least three tubes) number from five random fields in each well was counted and expressed as a percentage of branch points formed in the control sample, taken as 100%. As shown, TSA, MS275, SAHA and MC1568 HDACis strongly inhibit U87MG cells ability to form vascular tubes at 50 nM. (**C**) The cell rate viability assayed by MTT assay. At concentration of 50 nM, no significant differences are detected in treated groups as compared to the control group in 24 h. *** *p*-value < 0.001.

**Figure 4 cancers-11-00747-f004:**
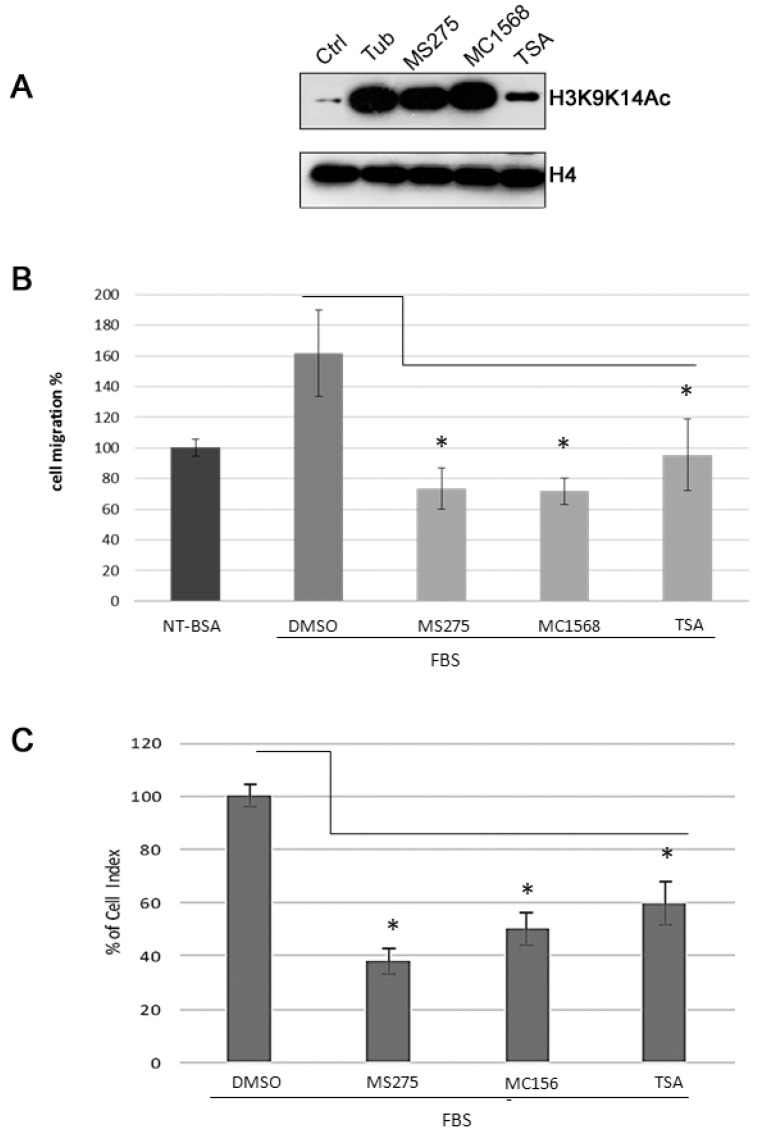
HDACis decrease U87MG directional cell migration towards FBS in two different transwell migration systems. (**A**) Representative western blot analysis of histonic cell extracts obtained from U87 MG cells treated with indicated compounds at 50 nM for 24 h. Immunoblotting was performed analysing the acetylated H3K9K14. H4 was used as loading control (**B**) Quantitative analysis of directional cell migration towards FBS using Boyden chamber migration assay. U87MG cells, suspended in serum-free media, were treated with 50 nM HDACis or DMSO 0.1% and were plated in the upper compartment of the Boyden chambers. 10% FBS-DMEM was added in the lower compartment as a chemoattractant. The chambers were incubated (37 °C, 5% CO_2_) and migrated cells were counted after 3 h. The number of migrated cells, in treated and control groups, is expressed as a percentage of the no treated migrated cells (NT) in the absence of chemo-attractant (basal migration), taken as 100%. (**C**) Quantitative migration analysis using xCELLigence^®^ Real-Time Cell Analyzer. U87 MG cells, suspended in serum-free media, were treated with the indicated compound at 50 nM and were plated into CIM-plate 16 with 8 μm pores; cells were seeded into the upper chamber with the lower chamber containing 10% FBS-DMEM as a chemoattractant. Experiments were carried out using the RTCA DP instrument, which was placed in a humidified incubator maintained at 37 °C with 5% CO_2_. The electronic sensors provided a continuous and quantitative measurement of the cell index (reflecting impedence, which depends on the number of attached cells) in each well. The CIM-plate 16 was monitored every 15 min for 24 h and the graph represents the percentage of cell index of treated groups respect to control group (DMSO), at 24 h. As shown, MS275, MC1568 and TSA significantly reduce the chemotactic response to FBS, in U87MG cells, as compared to cells migration in the presence of DMSO, the HDACis molecules solvent. Data show mean values from at least three independent measurements. Error bars show standard deviations. * *p*-value < 0.05.

**Figure 5 cancers-11-00747-f005:**
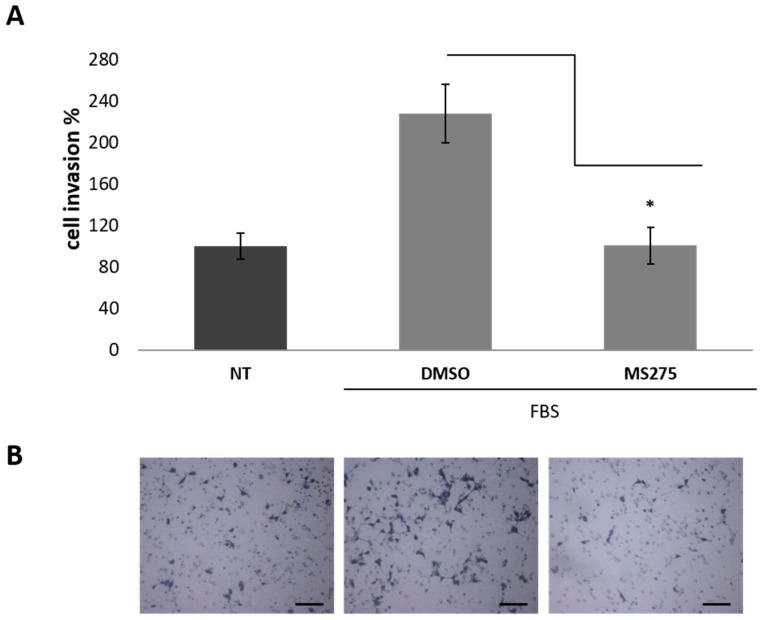
MS275 HDACi decreases U87MG Matrigel^®^ cell invasion. (**A**) Quantitative analysis of invasion Boyden chamber assay. At 50 nM MS275 reduces significantly the chemotactic response to FBS in U87MG cells, after 20 h, as compared to the control condition (DMSO 0.1%). Invaded cells were counted and expressed as a percentage of the no treated (NT) invaded cells in the absence of chemoattractant (random invasion), taken as 100%. Error bars show standard deviations. * *p*-value < 0.05. (**B**) Representative images of invaded cells on the microporous filters (20× magnification). Scale bars: 100 µm.

**Figure 6 cancers-11-00747-f006:**
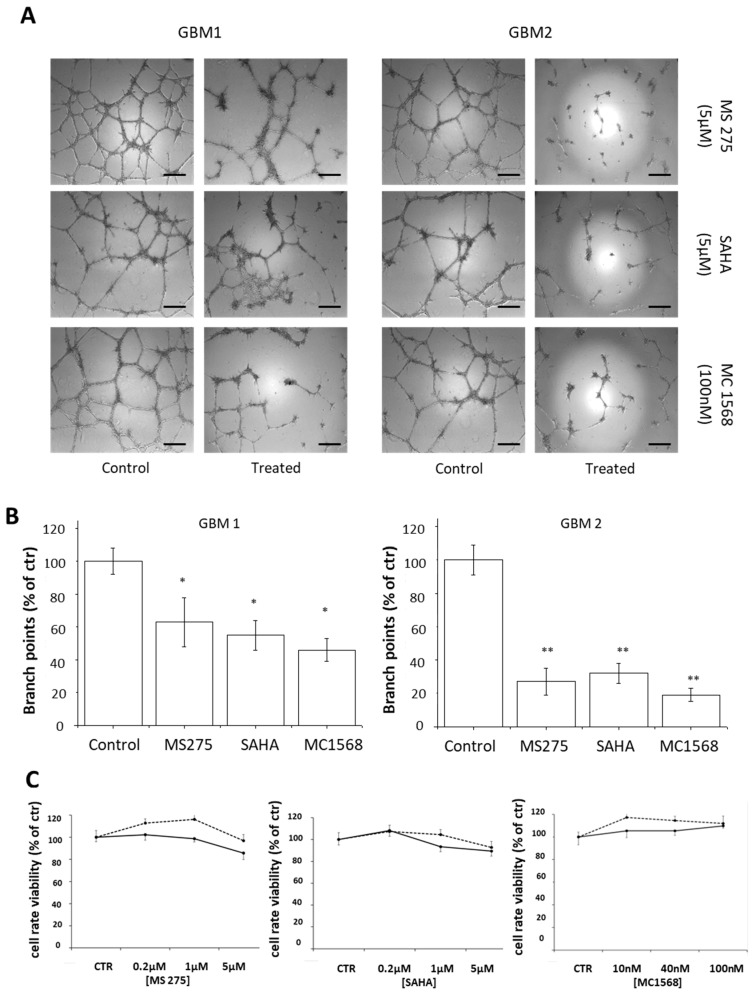
Effect of HDACis on the ability of CSCs isolated from two human GBMs to form vascular tubes. (**A**) Representative images of vascular tubes formed by GBM1 and GBM2 CSCs treated with vehicle (DMSO 0.1%) or with the tested drugs, at the indicated concentrations. Network of tubes, in each well, were directly analyzed under an inverted microscope with 5× phase contrast and imaged. Scale bar: 250 μm. (**B**) Quantification of vascular branching of A. Branch point number in each well was counted and expressed as a percentage of branch points formed in the control sample, taken as 100%. As shown, 24 h of treatment with MC1568, SAHA and MS275 reduce CSCs ability to form vascular tubes, with higher efficacy in GBM2 cells. (**C**) The cell viability rate, assayed by MTT reduction assay, evaluated in GBM CSCs; no significant differences are detected in treated groups as compared to the control group after 24 h of treatment. * *p*-value < 0.05, ** *p*-value < 0.01.

**Table 1 cancers-11-00747-t001:** Molecular characteristics of the histone deacetylase inhibitor (HDACis) used.

Group	Compound	Chemical Structure	HDAC Specificity
Hydroxamate	Vorinostat (SAHA)	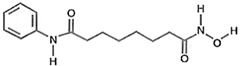	class I and II HDACs
	Trichostatin A (TSA)	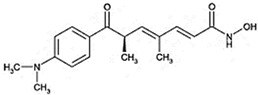	class I and II HDACs
	MC1568	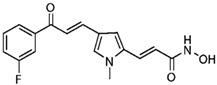	Class II HDACs
Benzamide	Entinostat (MS275)	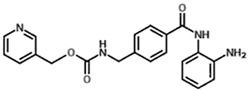	HDAC 1 and HDAC 3 (class I HDACs members)

**Table 2 cancers-11-00747-t002:** Characterization of CSCs source tumors.

Code	Age	Sex	WHO Grade	Histo Type	IHC	MIB-Index (%)	Cerebral Hemisphere/Lobe	Localization	Meningeal Infiltration
**Pt 1**	71	F	IV (primary, multicentric)	Neural	GFAP+	60	Left/T-P-O	Cortical-subcortical	no
**Pt 2**	40	F	IV (secondary, progression from oligodendroglioma)	Mesenchymal	GFAP + CD133+	30	Right/F-T	Cortical-subcortical	Yes

WHO: high-grade glioma

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
