# Peer review of "Histone Deacetylase Inhibitors Impair Vasculogenic Mimicry from Glioblastoma Cells"

_cancers, 2019, doi:10.3390/cancers11060747_

Round 1

Reviewer 1 Report

Authors revealed the anti-angiogenic efficacy of HDACi, in U87MG, rat glioma C6 cell line and patient-derived human GBM cancer stem cells. However, revisions should be done.

Scale bars are missing in figure 1A right panel and 1B, 2A, 3A, 5B and 6A. Also, indicate the magnification rate of figure 1, 5B in the figure legends.

Are they the same magnification between left and right panels of figure figure 1a?

Authors mentioned the side effect of conventional drugs including brain cells damages. However, there is no MTT assay result in HDACi treated normal brain cell lines. HDACis didn't showed any cytotoxicity against normal brain cell lines? 

What are the mechanisms of the anti-angiogenic effects of HDACis? Authors should reveal the mechanism of the effect.

What is the Institutional Ethical Committee approval number for using tumor specimens from patients. Also do you have the consent form from the patient about using their samples for this experiments?

Author Response

1.    Scale bars are missing in figure 1A right panel and 1B, 2A, 3A, 5B and 6A. Also, indicate the magnification rate of figure 1, 5B in the figure legends.Are they the same magnification between left and right panels of figure figure 1a?

We thank the Reviewer for his/her suggestions. The figures have been modified accordingly

2.    Authors mentioned the side effect of conventional drugs including brain cells damages. However, there is no MTT assay result in HDACi treated normal brain cell lines. HDACis didn't showed any cytotoxicity against normal brain cell lines? 

We want to thank the Reviewer for his/her helpful suggestion according to which we modified the manuscript. In particular, we have reported  studies showing a protective effect of the HDACis, including those used in our study such as SAHA and TSA, against neurodegeneration poiniting to them as neuroprotective agents (Annals of Clinical and Translational Neurology2015; 2(1): 79–101). This issue has been discussed in the revised version of the manuscript (introduction page 3, line 109-110)

3.    What are the mechanisms of the anti-angiogenic effects of HDACis? Authors should reveal the mechanism of the effect.

Molecular mechanisms underlying vasculogenic mimicry are largely unknown and identifying them could be the object of a large research project the is beyond the aim of this study. However, here we observed that  HDACis are able to impair also cell adhesion, invasion and migration that are all crucial events in the formation of new tube-like structures, in vitro as well as in vivo. Moreover, several  other studies report that HDACis are able to induce glioma cells differentiation and death via  epigenetic mechanisms still far to be completely elucidated. This issue has been extensively discussed in the manscript.

4.    What is the Institutional Ethical Committee approval number for using tumor specimens from patients. Also do you have the consent form from the patient about using their samples for this experiments?

Cancer stem cells have been obtained  in compliance with guidelines approved by Ethical Committee for animal use in cancer research at IRCCS Ospedale Policlinico San Martino (Genova, Italy) with number 17/12, approved on 14/09/2012. The consent form signed by patients about the use of  their samples for these experiments, are saved at IRCCS Ospedale Policlinico San Martino (Genova) and available upon request.  This issue has been added to the revised version of the manuscript in the method section at page 12, line 320.

Reviewer 2 Report

The manuscript submitted by Pastorino et al. entitled "Histone deacetylase inhibitors impair vasculogenic mimicry from glioblastoma cells" supports that HDACis inhibit human GBM cells ability to form vascular tubes. Giving a rationale for opening new therapeutic opportunities on the use of HDACis as agents  to target tumour neovascularization. The manuscript is well written, methods and results well described. However, there are some recommendations that need to be addressed as follows:

1. Authors tested Human GBM specimens and CSC cultures, however the data should be supported with animal studies, in order to better mimic the angiogenesis process and study therapeutic effect in vivo.

2. GBM specimens were obtained from patients whose underwent surgery and did not receive prior radio/chemo-therapy. Therefore, would be interest to also to include a group in in vitro assays treated with chemotherapy as comparison group to HDACis.

3. More human originating specimens could improve the quality and significance of the research as only samples from 2 patient were obtained. More cell lines should be tested as well.

4.  Quality of Figures, B,  C should be improved.

Author Response

1.     Authors tested Human GBM specimens and CSC cultures, however the data should be supported with animal studies, in order to better mimic the angiogenesis process and study therapeutic effect in vivo.

 We want thank the Reviewer for his/her helpful suggestion according to which we have modified the manuscript. In particular, we agree that studies in in vivo models  could better define the process of angiogenesis in humans, however this issue is beyond the aim of this article. We have discussed this limit of the research in the revised version of the manuscript (discussion page 10, line 244, 250)

2. GBM specimens were obtained from patients whose underwent surgery and did not receive prior radio/chemo-therapy. Therefore, would be interest to also to include a group in in vitro assays treated with chemotherapy as comparison group to HDACis.

 This is an interesting observation, however this could be the object of a further study. 

3. More human originating specimens could improve the quality and significance of the research as only samples from 2 patient were obtained. More cell lines should be tested as well.

 The fact that results obtained in this study are consistent in two glioblastoma cell lines from two different species (human and rat) and in CSCs from two patients with different histopathologic and genetic features, let us be confident that the effect is not restricted to a particular cell line and that these results could be reproduced also in other cell lines and CSCs. However, the limit of the number of sample used has been discussed in the revised version of the manuscript (discussion, page 11, line 301-302)

4.    Quality of Figures, B,  C should be improved.

The quality of the figure has been improved, as suggested by the Reviewer

Round 2

Reviewer 1 Report

Authors revised the manuscript well and it become better now. However, some more revisions should be done.

Please indicate the doses used at the figure legend or figure itself. I think I can't find the dose used in figure 4 and 5. Figure 6, upper figure 6A shows that dose of MS275 and SAHA is 5 μM and bottom figure 6A shows 5 mM for the same drugs. I think 5 μM is correct according to figure 6C.

Did HDACis inhibit their target HDACs with that doses? Author can reveal the anti-angiogenic mechanism of HDACis by showing the expression of target HDACs and at least vascular endothelial growth factor (VEGF) which is the main signaling pathway of angiogenesis.

There are two figures in figure 2, 3, 4, 5 and 6. I think authors intentionally added former version and revised version at the same time to compared them. But don't forget to remove former figures for final version.

Author Response

We want to thank the Reviewer for his/her helpful suggestions according to which we modified the manuscript.

 Did HDACis inhibit their target HDACs with that doses?

The activity of HDACis at the concentrations used in the experiments has been tested analyzing by western blotting the acetylated H3K9K14 in histonic extract from U87 exposed to the compounds. All the tested compounds showed to inhibit HDAC activity with subsequent hyper-acetylation of their targets. This result was described in the manuscript at Pag 4 line 129-131 as data not shown.

Author can reveal the anti-angiogenic mechanism of HDACis by showing the expression of target HDACs and at least vascular endothelial growth factor (VEGF) which is the main signaling pathway of angiogenesis.

HDACis inhibit HDAC function and, at the best of our knowledge, not their expression.

VEGF is the main stimulator of endothelial cell (ECs) proliferation and mobility during angiogenesis and high levels of VEGF are detected in the tumor microenvironment. However, vasculogenic mimicry, differently from angiogenesis, is a more complex process, involving several other mechanisms by which GBM cells can escape anti-angiogenic therapies  showing resistance to anti-VEGF drugs through the activation of alternative pathways. Thus,  vasculogenic mimicry is a VEGF-independent phenomenon.  For these reasons we did not evaluate the expression of VEGF but focalized our attention on cell properties such as migration, adhesion and invasion that are all crucial events in the formation of new tube-like structures in vitro as well as in vivo. This issue has been extensively addressed in the introduction (page 3, lines 80 to 91) and in the discussion of the manuscript (page 10, lines 286 to 295)

Round 3

Reviewer 1 Report

Authors should add the western blot analysis result of HDAC inhibition by each inhibitor.

Author Response

As suggested by the Reviewer, we added a representative western blotting analysis of histonic cell extracts obtained from U87MG cells treated with compounds that are representative of all the HDACis groups. The compounds were used at 50nM concentration as for the other tests  for 24 h. Immunoblotting was performed analysing the acetylated H3K9K14. H4 was used as loading control.  The manuscript as been modified in the figure 4, in the result section, page 7, lines 188-190 and the paragraph 4.7 has been added to the Mat and Met section.